# Causal Discovery for Linear Mixed Data

**Yan Zeng**                                                    YANAZENG013@TSINGHUA.EDU.CN
*Department of Computer Science and Technology, Tsinghua University, Beijing, 100084 China*

**Shohei Shimizu**                                         SHOHEI-SHIMIZU@BIWAKO.SHIGA-U.AC.JP
*Faculty of Data Science, Shiga University, Shiga, 522-8522 Japan*
*AIP Center, RIKEN, Tokyo, 103-0027 Japan*

**Hidetoshi Matsui**                                          HMATSUI@BIWAKO.SHIGA-U.AC.JP
*Faculty of Data Science, Shiga University, Shiga, 522-8522 Japan*

**Fuchun Sun**                                                       FCSUN@TSINGHUA.EDU.CN
*Department of Computer Science and Technology, Tsinghua University, Beijing, 100084 China*

**Editors:** Bernhard Schölkopf, Caroline Uhler and Kun Zhang

## Abstract

Discovery of causal relationships from observational data, especially from mixed data that consist of both continuous and discrete variables, is a fundamental yet challenging problem. Traditional methods focus on polishing the data type processing policy, which may lose data information. Compared with such methods, the constraint-based and score-based methods for mixed data derive certain conditional independence tests or score functions from the data's characteristics. However, they may return the Markov equivalence class due to the lack of identifiability guarantees, which may limit their applicability or hinder their interpretability of causal graphs. Thus, in this paper, based on the structural causal models of continuous and discrete variables, we provide sufficient identifiability conditions in bivariate as well as multivariate cases. We show that if the data follow our proposed restricted Linear Mixed causal model (LiM), such a model is identifiable. In addition, we proposed a two-step hybrid method to discover the causal structure for mixed data. Experiments on both synthetic and real-world data empirically demonstrate the identifiability and efficacy of our proposed LiM model.

**Keywords:** causal discovery, structural causal models, mixed data, identifiability

## 1. Introduction

Identifying the causal structure from purely observational data, termed as causal discovery, has been rapidly developed for the past decades with growing interest and has been widely applied in many domains (Pearl, 2000; Spirtes et al., 1993; Shimizu, 2014; Zhang and Hyvärinen, 2016). The traditional approaches to causal discovery roughly fall into two categories, namely constraint-based methods (Spirtes and Glymour, 1991; Spirtes et al., 1995), and score-based ones (Chickering, 2002). Since they may output Markov equivalence classes, i.e., a set of causal structures entailing the same conditional independences, they do not offer complete causal information. To distinguish different causal structures in the Markov equivalence class, several scholars derive additional assumptions on the data distribution and propose causal methods based on Structural Causal Models (SCM). These methods, including Linear Non-Gaussian Acyclic Models (LiNGAM) (Shimizu et al., 2006), Additive Nonlinear Models (ANM) (Hoyer et al., 2009), and Post Nonlinear (PNL) (Zhang and Hyvärinen, 2009), achieve the unique identifiability of the causal structure. Most existing causal

discovery methods focus on cases where the involved variables are either continuous or discrete only.

However, in many real-world scenarios such as economics (Wei et al., 2018), bioinformatics (Sedgewick et al., 2019), etc., the collected data often are a mixture of both continuous and discrete variables. When encountering such mixed data, one may ignore the discrete variables and apply the methods for continuous variables to estimate the partial causal network; or utilize a discretization policy to discretize the continuous variables, so that they can use those methods for discrete causal networks (Monti and Cooper, 1998; Chen et al., 2017). Both methods attempt to convert mixed data types into the same type, which is naive and can make it easy to lose data information, inducing non-negligible estimation errors. Yamayoshi et al. (2020) proposed a Latent-LiNGAM algorithm, which assumed that observed categorical variables are the result of discretizing latent continuous variables via link functions. But they did not concentrate on the model's theoretical analysis with mixed data. Apart from these methods, by and large, causal discovery algorithms for mixed data can be categorized into two classes: constraint-based and score-based ones. Constraint-based algorithms are those variants of PC (Pearl, 2000; Spirtes and Glymour, 1991), including Cui et al. (2016); Sedgewick et al. (2019); Tsagris et al. (2018), which cannot guarantee identifiability and are sensitive to samples.

Unlike the PC variants, score-based algorithms for mixed data do not use (un)conditional independence tests, but instead, optimize a likelihood score derived from mixed data's characteristics with the commonly-used greedy equivalence search framework. Such efforts include Li and Shimizu (2018); Huang et al. (2018); Andrews et al. (2019); Wei et al. (2018), etc. In particular, the first three efforts employ different score functions, i.e., with LiNGAM and the logistic regression model, regression model in RKHS, degenerate distributions, and respectively, whereas they do not provide the model's identifiability results and may return Markov equivalence classes. Wei et al. (2018) developed a mixed causal model and proved its identifiability in the bivariate cases. However, in their bivariate identifiability, they assumed the noises of continuous variables followed the Laplace distributions and the intercepts for discrete variables are zeros. Further, the bivariate identifiability is not qualified enough to handle multivariate cases whereas the multivariate data ordinarily exist in many applications.

Thus, in this paper, we propose a structural causal model that consists of both continuous and discrete variables, and give its sufficient identifiability conditions in bivariate as well as multivariate cases. Compared with the mixed model developed by Wei et al. (2018), we allow more non-Gaussian distributions to be followed by the noises of continuous variables and the intercepts for discrete variables are not restricted to be zeros. Further, we derive a two-step hybrid method to uniquely estimate the causal structure without discretization. In the first phase, we develop a log-likelihood score function to characterize the joint distribution for mixed data. It is optimized globally accompanied by the acyclicity and sparsity constraints. This global search however may stuck in local optima, which share the same skeleton with the ground truth structure [1]. To mitigate this issue, in the second phase, we search structures over the skeleton spaces and find the graph with the best score.

Our contributions mainly are detailed in two-fold:

---

1. We conjectured such local optima belong to the equivalence class of the global optima (the ground truth causal structure).

(i) For the mixed causal models that contain both continuous and discrete data, we prove the identifiability conditions in bivariate as well as multivariate cases. with which we enrich the identifiability space for causal discovery with mixed data.

(ii) We propose a score-based optimization method to infer the causal structure between mixed data. Experiments on synthetic and real-world data demonstrate our proposed method's efficacy, compared with other methods.

## 2. Preliminary and Model Definition

We consider linear mixed causal models. Speaking concretely, suppose we are given $p$ observed random variables, including discrete and continuous ones, i.e., $X = \{x_1, ..., x_p\}$. Since a categorical variable with $T$ categories can be regarded as $(T-1)$ binary variables, we assume that each discrete variable is binary (Wei et al., 2018). Further, we use the following assumptions from Li and Shimizu (2018):

A1. Observed variables $x_i$ ($i = 1, \ldots, p$) form a Directed Acyclic Graph (DAG).

A2. The value assigned to each continuous variable $x_i$ is a linear function of its parent variables denoted by $x_{\mathrm{pa}(i)}$ plus a non-Gaussian error term $e_i$, that is,

$$x_i = e_i + c_i + \sum_{j \in \mathrm{pa}(i)} b_{ij} x_j, \quad e_i \sim Non - Gaussian(\cdot), \tag{1}$$

where the error terms $e_i$ are continuous random variables with non-Gaussian densities, and the error variables $e_i$ are independent of each other. The coefficients $b_{ij}$ and intercepts $c_i$ are constants.

A3. For each discrete variable $x_i$, its value equals 1 if the linear function of its parent variables $x_{\mathrm{pa}(i)}$ plus a Logistic error term $e_i$ is larger than 0, otherwise, its value equals 0. That is,

$$x_i = \begin{cases} 1, & e_i + c_i + \sum_{j \in \mathrm{pa}(i)} b_{ij} x_j > 0 \\ 0, & \text{otherwise} \end{cases}, \quad e_i \sim Logistic(0, 1), \tag{2}$$

where the error terms $e_i$ are identical to those in Eq.(1), but follow the Logistic distribution.

**Definition 1 (Linear Mixed causal model, LiM)** *If a causal model for mixed data satisfies assumptions A1-A3, then this SCM is called a Linear Mixed causal model, abbreviated as LiM.*

Let $\mathcal{F} = \{f^{con}, f^{dis} | f^{con}(x, e) = bx + e + c, f^{dis}(x, e) = \begin{cases} 1, & bx + e + c > 0 \\ 0, & \text{otherwise} \end{cases} \}$ be a set of two functions which work on continuous and discrete variables, respectively, where $x, e, b$ and $c$ are a random variable of $X$, error term, coefficient and intercept, respectively. $\mathcal{P} = \{P^{con}, P^{dis}\}$ denotes the set of probabilistic distributions for continuous and discrete variables. Using these notations, our model can be rewritten as:

$$x_i = f_i(x_{\mathrm{pa}_i}, e_i), \quad e_i \sim P(e_i), \tag{3}$$

where $f_i \in \mathcal{F}$, and $P(e_i) \in \mathcal{P}$.

## 3. Identifiability Conditions of the LiM

Here we provide a sufficient identifiability condition for the LiM model, inspired by ideas of Wei et al. (2018) and Peters et al. (2014).

### 3.1. Bivariate cases

The LiM model of Section 2 is equivalent to the model of Wei et al. (2018) if the intercepts $c_i$ are taken to be zeros and the error terms $e_i$ follow the Laplace distributions $L(0, \alpha_i)$. Laplace distributions are commonly used in non-Gaussian models including independent component analysis and are known to be robust against the misspecification of the distributions if the right distribution is super-Gaussian (Hyvärinen et al., 2001). Wei et al. (2018) provided a sufficient identifiability condition for two-variable cases of their model, i.e., the two variables do not have the same marginal distributions if they are binary, all the probabilities and densities are positive, and the error variables are of non-zero variances. We show the identifiability of our model in a similar manner to Wei et al. (2018) [2]. The major difference lies in the fact that we allow more non-Gaussian distributions to be followed by the continuous error terms, rather than only the Laplace distributions.

Now we characterize the condition that the non-Gaussian distributions of our model need to satisfy.

**Condition 1** *The limit of non-Gaussian density ratio $\lambda$, defined as $\lambda := \lim_{x \to \pm\infty} \frac{P^{con}(x)}{P^{con}(x-b)}$, satisfies:*

$$\lambda = C, \tag{4}$$

*where $C$ is a non-zero finite constant. In other words, $\lambda$ is neither equal to zero nor infinity ($\lambda \neq 0, \infty$).*

It's noteworthy that basic non-Gaussian distributions satisfy Condition 1, including the Laplace distribution, Uniform distribution, Exponential distribution, and Gamma distribution, etc. With Condition 1 in the LiM model, we obtain our identifiability result in the bivariate case.

**Theorem 2** *Let the data $\boldsymbol{X} = \{x_i, x_j\}$ be generated by the LiM model in Eqs.(1)-(2) with Condition 1. Under the conditions that $x_i$ and $x_j$ do not share the same marginal distributions and $P(x_i = 1 \mid x_j = 0) = P(x_j = 1 \mid x_i = 0)$ holds [3] if they are both discrete, and all the probabilities are positive, the model is identifiable.*

**Proof** We prove the identifiability for the bivariate case from three aspects: i) both variables are continuous; ii) both variables are discrete; iii) one is continuous and the other is discrete.

i), if both variables are continuous, the model of Section 2 is a LiNGAM model (Shimizu et al., 2006). Therefore, the model is identifiable.

---

2. The idea that the model identifiability needs systematic differences in the marginal distributions of binary variables, can be found in other domains. For instance, in the domain of skewed latent variables, Wiedermann and von Eye (2020) showed that for the bivariate binary variables, due to the different marginal distributions between the outcome and predictor, asymmetry exists between causally competing non-hierarchical log-linear models.

3. Intuitively, this assumption implies $c_i = c_j$, which means the baselines are the same when predicting one discrete variable from the other. It can be tested easily by computing their values from the raw data. Note that the model of Wei et al. (2018) assumes $c_i = c_j = 0$, which is a special case of ours.

ii), suppose that two variables $\{x_i, x_j\}$ are binary. Assume that all the probabilities are positive and their marginal distributions are different. Then, we compare the following two models $x_j \to x_i$ and $x_i \to x_j$. The conditional probability $P(x_i \mid x_j)$ of the first model $x_j \to x_i$ is written as:

$$P(x_i = 1 \mid x_j) = \frac{1}{1 + e^{-(c_i + b_{ij}x_j)}}, \tag{5}$$

$$P(x_i = 0 \mid x_j) = 1 - P(x_i = 1 \mid x_j). \tag{6}$$

The conditional probability $P(x_j \mid x_i)$ of the second model $x_i \to x_j$ is written as:

$$P(x_j = 1 \mid x_i) = \frac{1}{1 + e^{-(c_j + b_{ji}x_i)}}, \tag{7}$$

$$P(x_j = 0 \mid x_i) = 1 - P(x_j = 1 \mid x_i). \tag{8}$$

Assume that two models give the same joint distribution of observed variables $x_i$ and $x_j$. Denote $P(x_i = 1)$ by $k_i$ and $P(x_j = 1)$ by $k_j$. Then,

$$k_i^{x_i}(1 - k_i)^{1-x_i}(\frac{1}{1 + e^{-(c_j + b_{ji}x_i)}})^{x_j}(1 - \frac{1}{1 + e^{-(c_j + b_{ji}x_i)}})^{1-x_j} \tag{9}$$

$$= k_j^{x_j}(1 - k_j)^{1-x_j}(\frac{1}{1 + e^{-(c_i + b_{ij}x_j)}})^{x_i}(1 - \frac{1}{1 + e^{-(c_i + b_{ij}x_j)}})^{1-x_i}. \tag{10}$$

If $x_i = x_j = 0$, we get $(1 - k_i)(1 - \frac{1}{1+e^{-(c_j)}}) = (1 - k_j)(1 - \frac{1}{1+e^{-(c_i)}})$. From $P(x_i = 1 \mid x_j = 0) = P(x_j = 1 \mid x_i = 0)$, we have $\frac{1}{1+e^{-(c_j)}} = \frac{1}{1+e^{-(c_i)}}$, which induces $k_i = k_j$. It contradicts with the assumption that the marginal distributions of $x_i$ and $x_j$ are different.

iii), suppose that one is continuous and the other is binary. Without loss of generality, assume that $x_i$ is continuous and $x_j$ is binary. We adopt the contradictory method, i.e., assuming two models in the following give the same joint distribution, then some conclusion would be drawn to contradict our model's conditions. Hence, we first compare such two models. The first model $x_i \to x_j$ is written as:

$$x_j = \begin{cases} 1, & e_j + c_j + b_{ji}x_i > 0 \\ 0, & \text{otherwise} \end{cases}, \quad e_j \sim Logistic(0, 1), \tag{11}$$

where $x_i = e_i \sim Non - Gaussian(\cdot)$. The second model $x_j \to x_i$ is written as:

$$x_i = e_i + c_i + b_{ij}x_j, e_i \sim Non - Gaussian(\cdot), \tag{12}$$

where $x_j = e_j \sim Logistic(0, 1)$. Assume that the two models give the same distribution of observed variables.

For the first model, the conditional probability of $x_j = 1$ given $x_i$ is given by

$$P(x_j = 1 \mid x_i) = \frac{1}{1 + e^{-(c_i + b_{ji}x_i)}}. \tag{13}$$

Then,

$$\lim_{x_i \to \infty} P(x_j = 1 \mid x_i) = \lim_{x_i \to \infty} \frac{1}{1 + e^{-(c_i + b_{ji}x_i)}} \tag{14}$$

$$= \begin{cases} 1 & (b_{ji} > 0) \\ 0 & (b_{ji} < 0) \end{cases}, \tag{15}$$

$$\lim_{x_i \to -\infty} P(x_j = 1 \mid x_i) = \lim_{x_i \to -\infty} \frac{1}{1 + e^{-(c_i + b_{ji} x_i)}} \tag{16}$$

$$= \begin{cases} 0 & (b_{ij} > 0) \\ 1 & (b_{ij} < 0) \end{cases}. \tag{17}$$

For the second model, the conditional probability of $x_j = 1$ given $x_i$ is given by

$$P(x_j = 1 \mid x_i) = \frac{P(x_j = 1, x_i)}{P(x_i)} \tag{18}$$

$$= \frac{P(x_i \mid x_j = 1)P(x_j = 1)}{P(x_i \mid x_j = 1)P(x_j = 1) + P(x_i \mid x_j = 0)P(x_j = 0)} \tag{19}$$

$$= \frac{P(x_j = 1)}{P(x_j = 1) + \frac{P(x_i \mid x_j = 0)}{P(x_i \mid x_j = 1)} P(x_j = 0)}. \tag{20}$$

Due to Condition 1, we obtain the limit of the density ratio as

$$\lim_{x_i \to \pm\infty} \frac{P(x_i \mid x_j = 0)}{P(x_i \mid x_j = 1)} = \lim_{x_i \to \pm\infty} \frac{P(x_i - c_i)}{P(x_i - c_i - e_i)} \tag{21}$$

$$= \lambda, \tag{22}$$

where $\lambda \neq 0$ and $\lambda \neq \infty$. Note that the limits $\lim_{x_i \to \pm\infty} P(x_j = 1 \mid x_i)$ under the second model are greater than 0 and smaller than 1 due to the assumption $P(x_j = 1) > 0$. This means that the limits $\lim_{x_i \to \pm\infty} P(x_j = 1 \mid x_i)$ under the second model are different from those of the first model, which contradicts the assumption that the two models give the same distribution of observed variables.

Thus, the model is bivariate identifiable if two variables $x_i, x_j$ do not have the same marginal distributions and $P(x_i = 1 \mid x_j = 0) = P(x_j = 1 \mid x_i = 0)$ holds in case that they are binary, all the probabilities and densities are positive, and the error variables are of non-zero variances. ∎

**Definition 3 (Bivariate Identifiable Set)** *Let $\mathcal{F} = \{f^{con}, f^{dis} \mid f^{con}(x, e) = bx + e + c, f^{dis}(x, e) = \begin{cases} 1, & bx + e + c > 0 \\ 0, & \text{otherwise} \end{cases}\}$ be a set of two functions which work on continuous and discrete variables, respectively. $\mathcal{P} = \{P^{con}, P^{dis}\}$ denotes the set of probabilistic distributions for continuous and discrete variables. Consider a mixed causal model with two variables $x_i$ and $x_j$, i.e., $x_j = e_j$ and $x_i = f_i(x_j, e_i)$ with $x_j \perp e_i$, where $\perp$ denotes the independence relation. We call a set $\mathcal{B} \subseteq \mathcal{F} \times \mathcal{P} \times \mathcal{P}$ as a bivariate identifiability set if the triple $(f_i, P(x_j), P(e_i))$ where $f \in \mathcal{F}$, and $P(x_j), P(e_i) \in \mathcal{P}$ hold, follows our LiM model's assumptions.*

Using the definition of the bivariate identifiable set $\mathcal{B}$, if the triple $(f_i, P(x_j), P(e_i))$ follows our LiM model, we have $(f_i, P(x_j), P(e_i)) \in \mathcal{B}$, which means that the bivariate mixed causal model is identifiable.

### 3.2. From bivariate to multivariate cases

Here we first review briefly the multivariate identifiablity of additive noise models for continuous random variables (Hoyer et al., 2009; Peters et al., 2014) and thereafter give our multivariate identifiablity of the LiM model for mixed data.

Intuitively, Peters et al. (2014) showed that under the assumption of causal minimality and that of positive densities, if two different additive noise graphs are assumed to give the same distribution of observed variables, it results in contradiction to the bivariate identifiability. They also pointed out that whenever they have a restriction that ensures identifiability in the bivariate case, the multivariate version remains valid. In fact, most parts of their proof other than the bivariate identifiability condition use only the general properties of non-parametric structural causal models with no hidden common causes and no cycles, and do not depend on the assumptions of additive noise models. Therefore, our LiM model also is identifiable for cases with more than two variable using the idea of Peters et al. (2014) based on the bivariate identifiability.

We then define the restricted mixed causal models to constrain conditional distributions and give identifiability analysis, in a similar manner to Peters et al. (2014).

**Definition 4 (Restricted LiM model)** *Consider a LiM model with $p$ variables. We call this SCM a restricted LiM model if for all $i \in \{1, ..., p\}$, $j \in \mathrm{pa}(i)$, and all sets $\mathbf{S} \subseteq \{1, ..., p\}$ with $\mathrm{pa}(i)\backslash\{j\} \subseteq \mathbf{S} \subseteq \mathrm{nd}(i)\backslash\{i, j\}$, there exists an $x_{\mathbf{S}}^*$ with $P_{\mathbf{S}}(x_{\mathbf{S}}^*) > 0$, s.t.*

$$(f_i(x_{pa(i)\backslash\{j\}}, \underbrace{\cdot}_{x_j}), P(x_j | x_{\mathbf{S}} = x_{\mathbf{S}}^*), P(e_i)) \tag{23}$$

*satisfies the assumptions and Condition 1 of the LiM's model in Section 2, i.e.,*

$$(f_i(x_{\mathrm{pa}(i)\backslash\{j\}}, \underbrace{\cdot}_{x_j}), P(x_j | x_{\mathbf{S}} = x_{\mathbf{S}}^*), P(e_i)) \in \mathcal{B}, \tag{24}$$

*where the underbrace with $x_j$ represents the input component of $f_i$ for the variable $x_j$, and $f_i \in \mathcal{F}$. $\mathrm{pa}(i)$ and $\mathrm{nd}(i)$ are index sets of $x_i$'s parents and non-descendants. Further, we require that the noise variables to have non-vanishing densities.*

**Theorem 5** *Let the data $\boldsymbol{X} = \{x_1, ..., x_p\}$ be generated by a restricted LiM model. Under the conditions that any two discrete variables do not share the same marginal distributions, all the probabilities are positive, and $P(\boldsymbol{X})$ satisfies the Markov and faithfulness conditions, the model is identifiable.*

**Proof** The theorem is proved by contradiction. We assume that our restricted mixed causal model is not identifiable, i.e., there exist two restricted mixed causal models $\mathcal{G}_1$ and $\mathcal{G}_2$, which induce the identical joint distribution $P(\mathbf{X})$. In such a case, we will show that $\mathcal{G}_1 = \mathcal{G}_2$ to induce the identifiability.

Consider two variables $x_i$ and $x_j$ in $\boldsymbol{X}$ where for the sets $\mathbf{Q} := \mathrm{pa}(i)^{\mathcal{G}_1}\backslash\{j\}$, $\mathbf{R} := \mathrm{pa}(j)^{\mathcal{G}_2}\backslash\{i\}$, and $\mathbf{S} := \mathbf{Q} \cup \mathbf{R}$, they satisfy i) $x_j \to x_i$ in $\mathcal{G}_1$ and $x_i \to x_j$ in $\mathcal{G}_2$; and ii) $\mathbf{S} \subseteq \mathrm{nd}(i)^{\mathcal{G}_1}\backslash\{j\}$ and $\mathbf{S} \subseteq \mathrm{nd}(j)^{\mathcal{G}_2}\backslash\{i\}$. Such two variables do exist (Peters et al., 2014). Firstly, due to ii) we get $e_i \perp (x_j, x_{\mathbf{S}})$ and $e_j \perp (x_i, x_{\mathbf{S}})$. Let $x_{\mathbf{S}}^* = \{x_{\mathrm{q}}, x_{\mathrm{r}}\}$. For the graph $\mathcal{G}_1$, we get $(f_i(x_{\mathrm{q}}, \cdot), P(x_j | x_{\mathbf{S}} = x_{\mathbf{S}}^*), P(e_i)) \in \mathcal{B}$, which satisfies the assumptions of our bivariate mixed causal model. It induces

$$x_i = f_i(x_{\mathrm{q}}, x_j^*), \quad x_{\mathrm{q}} \perp x_j^*, \tag{25}$$

where $x_j^* := x_j | x_{\mathbf{S}} = x_{\mathbf{S}^*}$ and $f_i \in \mathcal{F}$. For the graph $\mathcal{G}_2$, we get $(f_j(x_{\mathrm{r}}, \cdot), P(x_i | x_{\mathbf{S}} = x_{\mathbf{S}}^*), P(e_j)) \in \mathcal{B}$, which satisfies the assumptions of our bivariate LiM model. It induces

$$x_j = f_j(x_{\mathrm{r}}, x_i^*), \quad x_{\mathrm{r}} \perp x_i^*, \tag{26}$$

where $x_i^* := x_i | x_{\mathbf{S}} = x_{\mathbf{S}^*}$ and $f_j \in \mathcal{F}$. The above analysis contradicts the bivariate identifiability result in Theorem 2, hence we have $\mathcal{G}_1 = \mathcal{G}_2$. ∎

## 4. Optimization Method

To uncover the causal structure for mixed data that consist of both continuous and discrete variables, we propose an integrated hybrid score-based learning method. The objective function is based on the negative log-likelihood of the data. By instantiating the negative log-likelihood with the joint probability distribution of the mixed data, we get

$$
\begin{align}
\mathcal{L}(\mathbf{B}) &= -\log(P(\mathbf{X})) \tag{27} \\
&= -\log[\prod_t^n \prod_i^p P_d(x_{i,t} \mid x_{\mathrm{pa}(i),t})^{z_i} P_c(x_{i,t} \mid x_{\mathrm{pa}(i),t})^{1-z_i}] \tag{28} \\
&= -\sum_t^n \sum_i^p z_i \log[P_d(x_{i,t} \mid x_{\mathrm{pa}(i),t})] + (1 - z_i) \log[P_c(x_{i,t} \mid x_{\mathrm{pa}(i),t})], \tag{29} \\
&= -\sum_t^n \sum_i^p z_i \{x_{i,t} \log[\sigma(\mathbf{B})] + (1 - x_{i,t}) \log[1 - \sigma(\mathbf{B})]\} + \tag{30} \\
&\quad (1 - z_i) \log p_i(x_{i,t} - \sum_{k \in \mathrm{pa}(i)} b_{ik} x_{k,t}), \tag{31}
\end{align}
$$

where $\mathbf{B}$ is the adjacency matrix, and $x_{i,t}$ is the $t^{th}$ sample of the $i^{th}$ variable $x_i$. $n$ is the sample size. $P_d$ and $P_c$ denote the probability distribution of discrete and continuous variables, respectively. $z_i$ is an indicator variable, where $z_i = 1$ if $x_i$ is discrete while $z_i = 0$ otherwise. $\sigma(\mathbf{B}) = \frac{1}{1+e^{-(c_i + \sum_{l \in \mathrm{pa}(i)} b_{il} x_{l,t})}}$, while $p_i$ is the density function of the non-Gaussian error terms $e_i$ of continuous variables. In our method, we specified $p_i$ to be the density function of Laplace distributions. But, we can use other density functions as well. Thereafter, we seek to solve the following continuous optimization problem:

$$
\begin{align}
\min_{\mathbf{B}} \quad & \mathcal{L}(\mathbf{B}) + \Lambda ||\mathbf{B}||_1 \\
\text{subject to} \quad & h(\mathbf{B}) = 0,
\end{align} \tag{32}
$$

where $h(\mathbf{B})$ is an acyclicity constraint which ensures that $\mathbf{B}$ is a DAG (Zheng et al., 2018), $\Lambda$ is a regularization parameter, and $|| \cdot ||_1$ is an $l_1$ sparsity regularization. Following the optimization procedure in (Zeng et al., 2021b), we leverage the Quadratic Penalty Method (QPM) to estimate $\mathbf{B}$, converting Eq.(32) into an unconstrained function

$$
\min_{\mathbf{B}} \mathcal{S}(\mathbf{B}), \tag{33}
$$

where $\mathcal{S} = \mathcal{L}(\mathbf{B}) + \Lambda ||\mathbf{B}||_1 + \frac{\rho}{2} h(\mathbf{B})^2$ is the quadratic penalty function, and $\rho$ is a regularization parameter. Then we utilize the L-BFGS-B (Byrd et al., 1995) to solve Eq.(33). Due to the machine precision, it is well-known that it is hard for the estimated $b_{ij}$ to receive absolute zeros if such pairs have no edges (Zheng et al., 2018). Hence we give a small fixed threshold $\epsilon$ to rule out those whose estimated effects are lower than $\epsilon$.

---

**Algorithm 1** LiM Algorithm

---

**Require:** Data $X$; indicator vector $Z$; threshold $\epsilon$; tolerance parameter $\omega$.
**Ensure:** Connection strengths matrix $\mathbf{B}^*$.

    *Phase I: Global search*
  1: Optimize Eq.(32) to obtain $\hat{\mathbf{B}}$ using QPM with the tolerance parameter $\omega$,
  2: Rule out edges whose connection strengths are below $\epsilon$: $\hat{\mathbf{B}} = \hat{\mathbf{B}} \circ \mathbf{0}(\hat{b}_{ij} < \epsilon)$.
    *Phase II: Local search*
  3: Initiate a temporary minimum log-likelihood as $\mathcal{L}_{tmp}(\mathbf{B}^*) = \mathcal{L}(\hat{\mathbf{B}})$.
  4: **while** $\mathbf{B} \in \mathrm{Ske}(\hat{\mathbf{B}})$ **and** $h(\mathbf{B}) < \omega$ **do**
  5:     Compute the negative log-likelihood $\mathcal{L}(\mathbf{B})$ for $\mathbf{B}$.
  6:     **if** $\mathcal{L}(\mathbf{B}) < \mathcal{L}_{tmp}(\mathbf{B}^*)$ **then**
  7:         $\mathcal{L}_{tmp}(\mathbf{B}^*) = \mathcal{L}(\mathbf{B})$.
  8:     **end if**
  9: **end while**
10: **return** $\mathbf{B}^*$ with the minimal log-likelihood $\mathcal{L}_{tmp}(\mathbf{B}^*)$.

---

However, such an optimization method may easily stuck in local optima, which means the estimated skeleton is consistent with the ground-truth skeleton, while the estimated causal structure may not be consistent with the ground-truth one. To mitigate this issue, after obtaining the estimated adjacency matrix $\hat{\mathbf{B}}$, we tackle further the following combinatorial optimization problem:

$$\mathbf{B}^* = \arg \min_{\mathbf{B}\in\mathrm{Ske}(\hat{\mathbf{B}}),h(\mathbf{B})<\omega} \mathcal{L}(\mathbf{B}), \tag{34}$$

where $\mathrm{Ske}(\hat{\mathbf{B}})$ represents a set where the containing DAGs entail the same skeleton as $\hat{\mathbf{B}}$ and $\omega$ is a tolerance parameter. Compared with the traditional approaches that search over the discrete space of DAGs with the full graph, we perform our structure search over the narrowed space of DAGs within the estimated skeleton, which possesses an advantage in computational efficiency. The full algorithm is outlined in Algorithm 1.

As demonstrated in Algorithm 1, our LiM approach firstly performs global updates, estimating the connection strengths matrix $\hat{\mathbf{B}}$ in one step with continuous optimization techniques. Then, it performs a local update to search over the skeleton space, estimating one candidate DAG with one changing edge at each iteration in a combinatorial optimization manner. To conclude, the LiM approach is a two-step hybrid method, which takes advantages of both global and local search to avoid falling into local optima and to be more computationally efficient.

## 5. Experiments

In this section, we performed simulation experiments and employed our method to real-world application data to learn the causal graph with mixed data, evaluating the efficacy of our proposed method.

### 5.1. Synthetic data

To generate the data in simulations, we firstly established a randomly unweighted DAG according to the ER models (Zheng et al., 2018), where the number of edges was randomly selected. Given the

DAG, we assigned uniformly the edge weights from $[-2, -1] \bigcup [1, 2]$ to get an adjacency matrix **B**. Without loss of generality, the number of discrete variables was selected randomly from $[1, (p-1)]$, and thereafter we randomly assigned the discrete and continuous variables. Finally, the data were generated according to our LiM model in Eqs.(1)-(2).

We compared our method with a constraint-based method, a variant of PC algorithm (PC) (Spirtes and Glymour, 1991) as a representative. It discretized all continuous variables into discrete ones, following Li and Shimizu (2018). Besides, since PC may return a DAG pattern, PDAG, instead of a unique DAG, we took all possible instance for evaluation We compared with score-based methods, including the scores of Notears (Zheng et al., 2018) with the Logistic or Laplace distributions. We also compared with a commonly-used functional-based method, the LiNGAM method (Shimizu et al., 2006). To emphasize the necessity of our local search phase, we took our LiM method without the second phase as a comparison as well (mixed).

In these experiments, we evaluated the performance of all methods in terms of precision, recall and F1 score with both edges and causal directions, where the F1 score is defined as $\mathrm{F1} = \frac{2 \times \mathrm{Precision} \times \mathrm{Recall}}{\mathrm{Precision} + \mathrm{Recall}}$. For those continuous optimization methods, we chose the threshold $\epsilon = 0.1$, the tolerance $\omega = 1e - 8$, and the regularization parameter $\Lambda = 0.1$, while for those which exploit conditional independence tests, we fixed the significance level to be 0.01. For other parameters, we adopted their default settings.

We performed the simulations using i) different sample sizes, i.e., $n = 50, 100, 500, 1000,$ 2000, 5000, with bivariate and 5 mixed variables in turn. In addition, we generated the data with ii) different numbers of discrete variables ranging from 1 to $(p - 1)$ where each graph has 5000 samples, to test the robustness of our proposed LiM method. For each setting, we experimented with 50 realizations and reported the average results.

**Sensitivity to different sample sizes.** Figure 1 gives the results of the recovered causal graph with 2 or 5 mixed variables, compared with PC, logistic, laplace, LiNGAM, and mixed methods. The x-axis shows the sample sizes, while the y-axis is the recall, precision, or F1 score. Overall, our LiM method gives the best accuracy in both settings, which verified the identifiability results, especially in bivariate cases. More specifically, the LiM, and PC methods' accuracies increase remarkably along with the sample sizes up to 1000 in multi-variate causal networks. When the sample sizes reach 1000, it decreases slightly with increasing sample sizes. This phenomenon might be originated from the overfitting problem of optimization. Though our method is sensitive to sample sizes for learning causal networks, it does perform better than other comparisons. On the contrary, PC's unsatisfactory performance is basically due to the usage of conditional independence tests and the incapability of handling mixed data. Score-based methods are more robust to sample sizes compared with the constraint-based ones. However, since their score functions may not fit the mixed data or they may be trapped in local optima problems, their performances are not comparable despite stability.

**Sensitivity to different discrete variables.** Figure 2 reports the results of the recovered causal graph with different numbers of discrete variables ranging from 1 to $p-1$, where there are a total of 4 or 5 observed variables. The x-axis shows the number of discrete variables, while the y-axis is the recall, precision or F1 score. As shown, we can see that overall, our method and mixed method's performances tend to decrease distinctly as the number of discrete variables increases. The reason might be the violation of the assumption $P(x_i = 1 \mid x_j = 0) = P(x_j = 1 \mid x_i = 0)$ and with the increasing number of discrete variables, such an assumption is more likely to be violated. But our

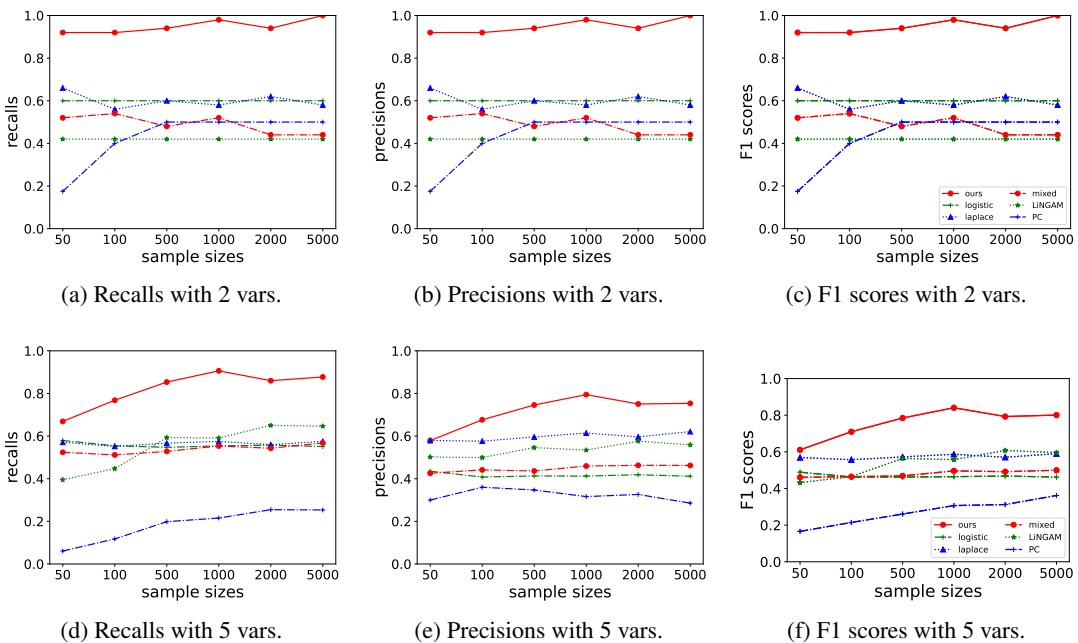

(a) Recalls with 2 vars.     (b) Precisions with 2 vars.     (c) F1 scores with 2 vars.

(d) Recalls with 5 vars.     (e) Precisions with 5 vars.     (f) F1 scores with 5 vars.

Figure 1: Recalls, precisions, and F1 scores of recovered causal graphs for bivariate and 5 mixed variables with different sample sizes. Higher F1 scores, recalls, and Precisions mean higher accuracies.

method performs overall better than other comparisons, indicating the capability of handling mixed data.

## 5.2. Real-world data

*Boston housing data set.* We then applied our LiM method to a real-world Boston housing data set, which was collected at the UCI Repository (Dua and Graff, 2017). Such a data set contains 506 data points and we chose 11 variables for the experiments, where the chosen continuous variables are identical to Zhang et al. (2011) and the only binary variable is included. They are *CRI* (continuous, per capita crime rate by town), *IND* (proportion of non-retail business acres per town), *CHA* (binary, if tract bounds river or not), *NOX* (continuous, nitric oxides concentration), *RM* (continuous, average number of rooms per dwelling), *AGE* (continuous, proportion of owner-occupied units built prior to 1940), *DIS* (continuous, weighted distances to five Boston employment centres), *TAX* (continuous, full-value property-tax rate per 10,000 dollars), *B* (continuous, the proportion of blacks by town), *LST* (continuous, percentage lower status of the population), and *MED* (continuous, median value of owner-occupied homes). We used the same settings as in the simulation experiments, i.e., we set the ruled-out threshold $\epsilon = 0.1$ and tolerance parameter $\omega = 1e - 8$. To provide reliable performance, due to the different scales of the large number of variables, we standardized the continuous variables before employing our method. The resulting causal graphs are demonstrated in Figure 3. Since the logistic method estimated more than 40 edges, and LiNGAM estimated nearly 20 edges, which have much more spurious edges than others, we only showed the comparison results of laplace and PC methods. As shown in Figure 3(a), on the one hand, though it is arguable that *RM* may not

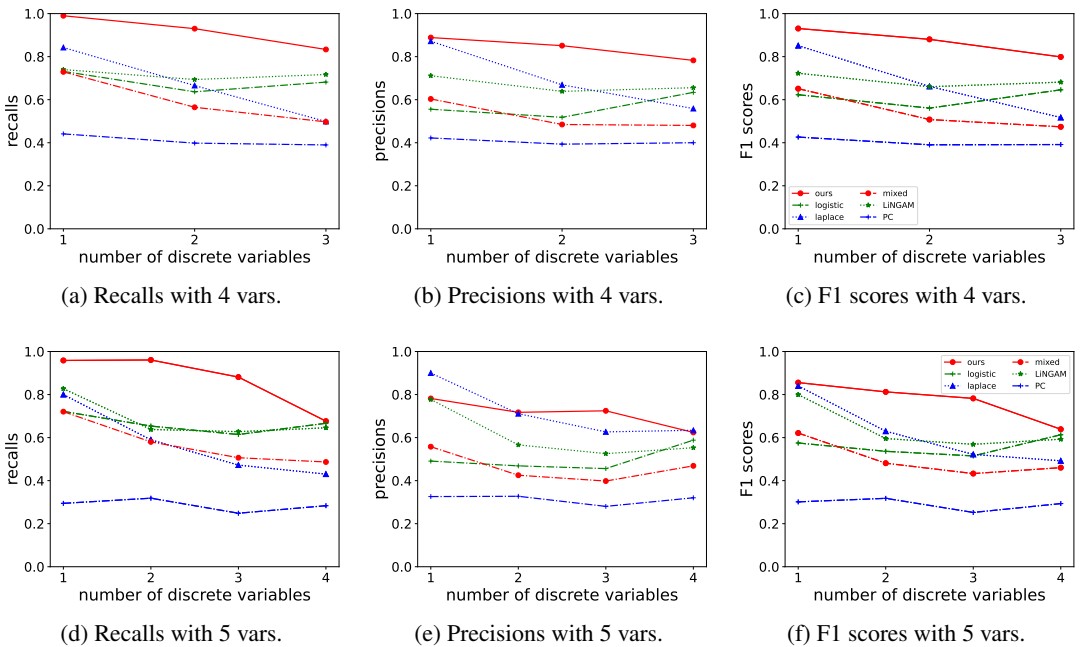

(a) Recalls with 4 vars.     (b) Precisions with 4 vars.     (c) F1 scores with 4 vars.

(d) Recalls with 5 vars.     (e) Precisions with 5 vars.     (f) F1 scores with 5 vars.

Figure 2: Recalls, Precisions, and F1 scores of recovered causal graphs with different numbers of discrete variables, where the sample size is 5000.

be an effect variable, we still found some interesting conclusions which were accordance with our common understandings. For example, *MED* is influenced by *LST*, which is determined by some house-related indicators, i.e., *IND*, *AGE* and *TAX*; there is no direct link between *NOX* and *MED* but they are dependent through intermediate causal relationships (Margaritis, 2005); it is reasonable that *TAX*, which reflects the government's housing policy, influences *IND*, *LST*, and *CRI* (Kenyon et al., 2012). On the other hand, laplace and PC methods estimated the TAX as an effect variable, which was not consistent with our common understanding (Kenyon et al., 2012). Overall, the results illustrated the effectiveness of our proposed LiM method in inferring causal graphs from mixed data.

## 6. Conclusions

In this paper, we provided complete identifiability conditions for causal discovery with linear mixed data that consist of continuous and discrete variables, both in bivariate and multivariate cases. Further, we proposed a two-step hybrid approach to uniquely identify the causal structure. Experiments on synthetic as well as real-world data demonstrate that our LiM method outperformed the comparisons.

There are several questions that we aim to answer in the future research. First, in this paper, we give mathematical identifiability for mixed data under the causal sufficiency. But in many realistic applications, causal sufficiency may not hold and there may exist latent variables or confounders in the underlying causal graphs. Thus, it is significant to develop a general identifiability condition without the causal sufficiency for mixed data. Second, the linearity assumption may decrease the

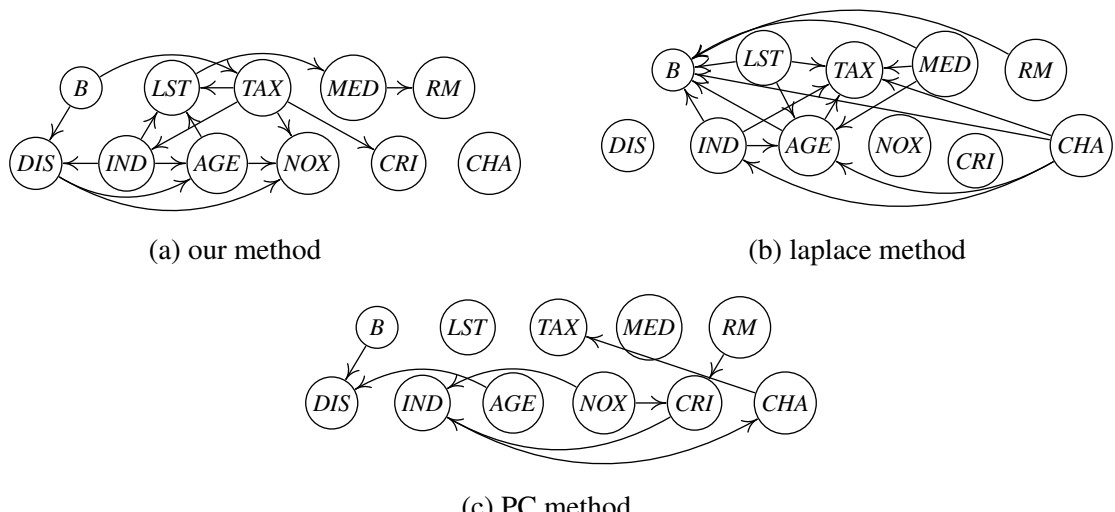

(a) our method

(b) laplace method

(c) PC method

Figure 3: Results of (a) our LiM method, (b) laplace method and (c) PC method applied to the Boston housing data set.

learning effect and limit the applicability. It is essential to generalize the identifiability to cases where there are nonlinear relationships, not only for bivarate but also multi-variate causal networks. As for the nonlinear causal modeling, there are some aspects that we could consider deeper research. For instance, we could focus on the cases where the mixed data are generated by the PNL (Zhang and Hyvärinen, 2009), or where they are generated by a nonlinear deterministic relation without noise (Janzing et al., 2012; Zeng et al., 2021a), etc. Third, in practice, there may also exist other types of discrete variables, e.g., multi-categorical variables or ordinal variables. Ignoring ordinal information may have adverse effects on model estimation (von Eye and Wiedermann, 2018). Thus, it is desirable to extend our method to cover general data types.

## Acknowledgments

We are grateful to the anonymous reviewers whose careful comments and suggestions helped improve this manuscript. This work was supported by the Grants ONR N00014-20-1-2501 and KAK-ENHI 20K11708.

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
