# OpenReview forum: "Causal Discovery for Linear Mixed Data"
_cclear.cc/CLeaR/2022/Conference — CLeaR 2022 Poster_

### Official Review · Reviewer_Gufh · 2021-11-19

**Confidence:** 3
**Overall Score:** 7

**Main Review:**

The paper introduces a novel approach to discover causal structures in mixed (continuous/categorical) data. Such data occur across all empirical sciences. Thus, the presented approach can be considered of high relevance for the audience of causal machine learning and beyond. However, there is a small number of aspects (summarized below) that requires additional attention prior to recommending acceptance of the manuscript in the Proceeding of Machine Learning Research.

First, the paper is well organized. However, in general, the paper might benefit from careful proofreading of a professional editor. Some parts of the paper are a bit hard to follow which may mitigate clarity.

p. 2, 2nd paragraph:  For sake of completeness, I suggest including another mixed data causal discovery algorithm proposed by Yamayoshi et al. (2020). These authors proposed a Latent-LiNGAM (L-LiNGAM) algorithm. Here, it is assumed that observed categorical variables are the result of discretizing latent continuous variables via link functions (in case of the identity link L-LiNGAM reduces to the classic LiNGAM). The paper would benefit from incorporating this previous work. Also, I recommend including this algorithm in the artificial data experiment as a benchmark.

p. 3, last paragraph: Similar to Wei et al. (2018), the presented algorithm makes use of the fact that causal models are identifiable due to systematic differences in the marginal distributions of binary variables. It may be worth noting that a similar result has already been discussed by Wiedermann and von Eye (2017) in the domain of skewed latent variables. Following the idea of Dodge and Rousson (2000) – who showed that, when the error is Gaussian, the skewness of the true outcome variable will always be smaller than the skewness of the true predictor –these authors showed that, for the bivariate case, asymmetry exists between causally competing non-hierarchical log-linear models (i.e., models without one of the two main effects). Because the marginal distribution of the true outcome will be closer to symmetry than the marginal distribution of the predictor, causal model selection can be performed.

p. 10, Section 5.1 (Synthetic data): The selection of competing algorithms seems somewhat suboptimal. For example, why was LiNGAM included in the simulation? Since this algorithm has been designed for continuous variables only, it does not serve as a reliable benchmark. Instead, I suggest including previously suggested algorithms for mixed data (e.g., the L-LiNGAM approach mentioned above).

p. 10: Please provide a citation for the NOTEARS algorithm.

p. 10 & 11 (Results section): While the accuracy results for the proposed approach are impressive and convincing, I do not fully agree with some of the write-up in the corresponding results section. For example, the authors state that accuracies of the proposed approach and PC “…increase remarkably along with the sample sizes”, this statement only holds for sample sizes up to 1000. Beyond n = 1000, it seems that the accuracy of the proposed approach slightly decreases again. Here, the authors should clarify whether this is an ignorable instability due to the selected number of iterations, or whether this is a phenomenon inherent to the approach. Also, please note that the performance of the PC algorithm seems to be stable for the bivariate case which contradicts the authors’ statement cited above. Finally, when discussing results of varying the number of discrete variables, it may be worth noting that the accuracy of the proposed method tends to decrease, suggesting that the accuracy may be affected by the number of categorical variables.

p. 12 (Conclusion section): I fully agree that future work needs to be done with respect to nonlinear causal relations. However, given that the authors state that identification results also hold for the ANM, it seems that this work has already been done. Thus, I suggest being a little more specific. Which aspects of causally modeling nonlinear relations requires future attention? Finally, another important future direction may be the inclusion of categorical variables with more than two categories. Currently, the authors make use of the fact that multi-categorical variables can be represented by k – 1 binary variables. However, such an approach might be suboptimal, in particular, in the presence of ordinal variables. It is well-known that ignoring ordinal information can have adverse effects on model estimation (see von Eye and Wiedermann, 2018, for a recent illustration). The paper might benefit from incorporating these perspectives.

References

Dodge, Y., & Rousson, V. (2000). Direction dependence in a regression line. Communications in Statistics-Theory and Methods, 29(9–10), 1957–1972. https://doi.org/10.1080/03610920008832589

von Eye, A., & Wiedermann, W. (2018). Strengthening arguments based on scale levels? Journal for Person-Oriented Research, 4(1), 45-47.

Wiedermann, W., & von Eye, A. (2017). Log-linear models to evaluate direction of effect in binary variables. Statistical Papers. (in press).


**Summary:**

The main contribution of the paper is an introduction into a novel causal discovery algorithm for mixed (i.e., continuous and categorical) data. In the present approach, continuous variables are assumed to deviate from the Gaussian distribution and categorical variables are assumed to be the result of discretization of a latent variable that follows a logistic distribution. The authors show that the presented approach enables one to uncover causal structures beyond Markov equivalent classes in both, bivariate and multivariate data settings. A two-step hybrid approach is presented that, in the first step, estimates the log-likelihood score function for the joint distribution of mixed data, and, in the second step, identifies the causal graph with the best score value among structures of the skeleton space. The performance of the algorithm is evaluated using artificial data experiments. Here, simulation results suggest that the presented approach outperforms five competing algorithms (2 variants of the NOTEARS algorithm considering logistics and Laplace distributions, the PC algorithm, LiNGAM originally proposed for continuous variables only, and a second variant of the proposed algorithm without the second search phase). The Boston housing dataset is used to illustrate the presented approach in practice. Here, the results are mixed. Some of the identified paths come with high face validity, other paths are more likely to be spurious due to logical inconsistencies.

---

> ### Author Response · Authors · 2021-12-04
> **Response to Reviewer Gufh (Part I)**
>
> We would like to thank the reviewer Gufh for the inspiring feedback and useful comments. Please see below our responses.
>
> Q1: p. 2, 2nd paragraph: For sake of completeness, I suggest including another mixed data causal discovery algorithm proposed by Yamayoshi et al. (2020). These authors proposed a Latent-LiNGAM (L-LiNGAM) algorithm. Here, it is assumed that observed categorical variables are the result of discretizing latent continuous variables via link functions (in case of the identity link L-LiNGAM reduces to the classic LiNGAM). The paper would benefit from incorporating this previous work. Also, I recommend including this algorithm in the artificial data experiment as a benchmark.
>
> A1: Thanks a lot for your valuable comments. The major difference between both methods is that they (Yamayoshi et al. 2020) do not concentrate on the theoretical analysis of the model with mixed data but the estimation method. However, our work aims at giving identifiability results for the mixed model and proposing an estimation method. We will add this work as a reference. Also, we could not find the available code online up until now. If we obtain the code, we will be ready to include it as a benchmark in the simulation.
>
> Q2: p. 3, last paragraph: Similar to Wei et al. (2018), the presented algorithm makes use of the fact that causal models are identifiable due to systematic differences in the marginal distributions of binary variables. It may be worth noting that a similar result has already been discussed by Wiedermann and von Eye (2017) in the domain of skewed latent variables. Following the idea of Dodge and Rousson (2000) – who showed that, when the error is Gaussian, the skewness of the true outcome variable will always be smaller than the skewness of the true predictor –these authors showed that, for the bivariate case, asymmetry exists between causally competing non-hierarchical log-linear models (i.e., models without one of the two main effects). Because the marginal distribution of the true outcome will be closer to symmetry than the marginal distribution of the predictor, causal model selection can be performed.
>
> A2: Thanks so much for your information. We will mention these works for bivariate cases.
>
> Q3: p. 10: Please provide a citation for the NOTEARS algorithm.
>
> A3: We will provide a citation for the NOTEARS algorithm.
>
> Q4: p. 10 & 11 (Results section): While the accuracy results for the proposed approach are impressive and convincing, I do not fully agree with some of the write-up in the corresponding results section. For example, the authors state that accuracies of the proposed approach and PC “…increase remarkably along with the sample sizes”, this statement only holds for sample sizes up to 1000. Beyond n = 1000, it seems that the accuracy of the proposed approach slightly decreases again. Here, the authors should clarify whether this is an ignorable instability due to the selected number of iterations, or whether this is a phenomenon inherent to the approach. Also, please note that the performance of the PC algorithm seems to be stable for the bivariate case which contradicts the authors’ statement cited above. Finally, when discussing results of varying the number of discrete variables, it may be worth noting that the accuracy of the proposed method tends to decrease, suggesting that the accuracy may be affected by the number of categorical variables.
>
> A4: We will rephrase the experimental analysis on the sensitivity to the sample sizes. As shown in Fig. 1, our proposed method increases remarkably along with the sample sizes up to 1000. When the sample size reaches 1000, it looks to decrease slightly with increasing sample sizes. This phenomenon might be originated from the overfitting problem of optimization. Though our method is sensitive to sample sizes for learning causal networks, it does perform better than other comparisons. As for the effects of the number of categorical variables, we performed experiments on a causal network with 5 variables and 5000 sample sizes. A part of the result is presented below.
>
> |No. of dis. var.|PC|LiNGAM|mixed|laplace|logistic|ours|
> |-|-|-|-|-|-|-|
> |1	|0.330229	|0.799588	|0.621185	|0.840026	|0.574885	|$\textbf{0.855555}$|
> |2	|0.362768	|0.595599	|0.480997	|0.629606	|0.536318	|$\textbf{0.812429}$|
> |3	|0.340048	|0.568806	|0.454009	|0.521855	|0.514909	|$\textbf{0.782167}$|
> |4	|0.299163	|0.592565	|0.432868	|0.492097	|0.613164	|$\textbf{0.638836}$|
>
> We see that the proposed method’s performance tends to decrease as the number of discrete variables increases. But our method overall performs better than other comparisons. We will add this description in Section 5.1.

---

> > ### Author Response · Authors · 2021-12-04
> > **Response to Reviewer Gufh (Part II)**
> >
> > Q5: p. 12 (Conclusion section): I fully agree that future work needs to be done with respect to nonlinear causal relations. However, given that the authors state that identification results also hold for the ANM, it seems that this work has already been done. Thus, I suggest being a little more specific. Which aspects of causally modeling nonlinear relations requires future attention? Finally, another important future direction may be the inclusion of categorical variables with more than two categories. Currently, the authors make use of the fact that multi-categorical variables can be represented by k – 1 binary variables. However, such an approach might be suboptimal, in particular, in the presence of ordinal variables. It is well-known that ignoring ordinal information can have adverse effects on model estimation (see von Eye and Wiedermann, 2018, for a recent illustration). The paper might benefit from incorporating these perspectives.
> >
> > A5: We agree with you. We will add the inclusion of categorical variables as one of the future lines. As for the nonlinear causal modeling, there are some aspects that we could consider deeper research. For instance, we could focus on the cases where for the mixed data, continuous variables are generated by the Post Non-Linear Model (PNL, it has a very general form about the data mechanism); or where continuous variables are generated by a nonlinear deterministic relation where $x_i = f(x_{pa(i)})$ without noise; or where the discrete variable is generated by a mechanism where its value equals 1 if the nonlinear function of its parent variables plus an error is larger than 0, otherwise, its value equals 0, etc. These cases have much more challenges than we might not imagine. Thus, they are worthwhile being explored. Thank you for your valuable suggestions!

---

### Official Review · Reviewer_eCeE · 2021-11-21

**Confidence:** 4
**Overall Score:** 6

**Main Review:**

**Update**: After reading the other reviews and discussing with the authors, I have increased my score from 5 to 6.

The authors build upon the work of Shimizu et al. (2006) for handling continuous variables, and on the work of Wei et al. (2018) for handling discrete variables. The extension over Wei's approach is that the LiM model can incorporate more types of non-Gaussian distributions for the continuous variables. However, the method's applicability is severely limited by the causal sufficiency assumption, which will not hold for most realistic settings. I would have liked to have seen this limitiation discussed more explicitly in the paper.

I think the paper could be significantly improved in terms of clarity. The paper is unfortunately pockmarked with writing errors, which hinder readability. Moreover, there are parts such as Corollary 1 whose purpose in the text is unclear to me. The authors require Condition 1 on the tails of the distribution to prove Theorem 2, but I am not sure why the condition of having a finite non-zero limit needs to be separated in the two cases S1 and S2. Conversely, some parts are not explained in enough detail, as far as I am concerned. For instance, the authors claim, if I understand correctly, that by setting $x_i = x_j = 0$ in equations (9) and (10), which are actually supposed to be a single equation, we conclude that $k_i = k_j$. However, what we get is that $\frac{1-k_i}{1 - e^{-c_i}} = \frac{1-k_j}{1 - e^{-c_j}}$. It seems to me that some non-obvious steps are missing there.

On page 2, what do you mean by "The output causal structure here may fall into the solutions up to the ground truth's skeleton"? Similarly, on page 9, what do you mean by "an optimization method may easily fall into its skeleton solutions" and what is the "sub-optimality" you are referring to?

On page 7, the authors relate their reasoning to the proof of Theorem 28 in Peters et al. (2014). I do not think it is necessary to copy the text of Remark 30 from Peters et al. (2014), especially since it is hard to understand without context.

On page 10, in the Experiments section, the authors mention that they take the instance of a PDAG returned by PC for evaluation. Which instance do they use exactly? Depending on how the evaluation is performed, one instance might perform better than another. It is also not exactly clear to me how the performance is being evaluated. Are the precision, recall, and F1 score computed in terms of the skeleton, that is, having the corrected edges, or do you also consider the edgemarks? When is a discovery correct and when is it wrong?

Overall, I think that while the LiM model may improve over the state-of-the-art, albeit incrementally, the paper lacks clarity and rigor, and is need of some serious rewriting before it is ready for submission.

Typos, writing errors, and other comments:
- In the introduction, the categorization of causal discovery algorithms into constraint-based and score-based is introduced twice (on page 1, and then again at the beginning of page 2).
- Page 1: "may output Markov equivalence class" -> "output Markov equivalence classes"; "same conditional independence" -> "same conditional independences"; "These methods including" -> "These methods, including"; "on cases when" -> "on cases where"
- Page 2: "which are naive and are easy to lose data information" -> "which is naive and can make it easy to lose data information"; "can not" -> "cannot"; "may return Markov equivalence class" -> "may return Markov equivalence classes"; "accompanied with" -> "accompanied by"; "ratio by" -> "ratio of"; "T class" -> "T categories"; "And further" -> "Further"
- Page 3: "equals to 1" -> "equals 1"; "equals to 0" -> "equals 0"; "the assumptions A1-A3" -> "assumptions A1-A3"
- Page 4: "condition probability" -> "conditional probability"
- Page 5, Equation (17): $b_{ij}$ should be $b_{ji}$ for both branches
- Page 6: "Due to the Condition 1" -> "Due to Condition 1"
- Page 6, Definition 3: The authors use the notation $\perp$ presumably to denote (conditional) independence, but do not explain what it means.
- Page 7, Definition 4: It would be nice to introduce the sets **pa** and especially **nd** for those who are not so familiar with the notation. What do you mean by "$x_j$'s component with $f_i$" when explaining the underbrace?
- Page 7, Theorem 5: "faithful" -> "faithfulness"
- Page 9: "that the estimated $b_{ij}$ is hard to receive absolute zeros" -> "that it is hard for the estimated $b_{ij}$ to receive absolute zeros";
- Page 10: "Logistic (logistic)" -> "logistic"; "Laplace (laplace)" -> "Laplace"; "Re." -> "recall" & "Pre." -> "precision" (The abbreviations seem completely unnecessary.); "to the sample sizes" -> "to sample sizes"
- Page 12: "lower status of the population" -> "percentage lower status of the population"; "may not an effect variable" -> "may not be an effect variable"; The variable INDUS is later referred to as IND, which is inconsistent.
- Page 14: "form" should be "from"

**Summary:**

The authors propose a method for causal discovery from mixed data under the linearity and causal sufficiency assumptions. The proposed method relies on assuming non-Gaussian error terms for continuous variables and logistic error terms for discrete variables. The authors show that, under certain conditions, the generating causal DAG in their Linear Mixed causal model (LiM) is fully identifiable. Finally, they showcase the performance of their LiM model on a number of simulations and in a real-world experiment on the Boston housing dataset.

---

> ### Author Response · Authors · 2021-12-04
> **Response to Reviewer eCeE (Part I)**
>
> We would like to thank the reviewer eCeE for the thoughtful feedback and constructive comments. Please see below our responses.
>
> Q1: the method's applicability is severely limited by the causal sufficiency assumption, which will not hold for most realistic settings. I would have liked to have seen this limitation discussed more explicitly in the paper.
>
> A1: Thanks for pointing out this issue. Causal discovery without the causal sufficiency assumption is still an interesting and open challenge, especially when the observed data are mixed with both continuous and discrete variables. When the observed data are mixed with both continuous and discrete variables, to our best knowledge, though there is some progress on this challenge [1], no mathematical proof of the identifiability for multivariate cases has been provided even under the causal sufficiency. Therefore, we believe that it is an important contribution to first give proof of the identifiability under the causal sufficiency. We will discuss it more in Section 6 Conclusions.
> [1] Raghu, V.K., Ramsey, J.D., Morris, A. et al. Comparison of strategies for scalable causal discovery of latent variable models from mixed data. Int J Data Sci Anal 6, 33–45 (2018). https://doi.org/10.1007/s41060-018-0104-3
>
> Q2: There are parts such as Corollary 1 whose purpose in the text is unclear to me. The authors require Condition 1 on the tails of the distribution to prove Theorem 2, but I am not sure why the condition of having a finite non-zero limit needs to be separated in the two cases S1 and S2.
>
> A2: Corollary 1 is designed to help demonstrate in detail that the widely-used non-Gaussian distributions could satisfy Condition 1. We believed that Corollary 1 will promote the interpretation of Condition 1.
>
> Q3: The authors claim, if I understand correctly, that by setting $xi=xj=0$ in equations (9) and (10), which are actually supposed to be a single equation, we conclude that $ki=kj$. However, what we get is that $\frac{1−k_i }{1−e^{-c_i}}=\frac{1−k_j}{1−e^{-c_j}}$. It seems to me that some non-obvious steps are missing there.
>
> A3: You are correct. Thanks for your excellent comment! When setting $xi=xj=0$, we obtain $(1-k_i)(1-\frac{1}{1+e^{-c_j}}) = (1-k_j)(1-\frac{1}{1+e^{-c_i}})$ from Eq.(9) and (10) shown below. Then we introduce another assumption, i.e., $p(x_j=1|x_i=0) = p(x_i=1|x_j=0)$, which implies $\frac{1}{1+e^{-c_j}} = \frac{1}{1+e^{-c_i}}$ holds and thus $c_i=c_j$ holds. With $c_i=c_j$, it can be induced that $ k_i = k_j$ (Please note that the model of Wei et al. (2018) assumes $c_i=c_j=0$, which is a more restrict assumption than ours). In this case, Wei’s model (Wei et al., 2018) is a special case of our model. Intuitively, this assumption “ $p(x_j=1|x_i=0) = p(x_i=1|x_j=0)$” means the baselines are the same when predicting one discrete variable from the other. It can be tested easily by computing their values from the raw data. Besides, it is worth mentioning that our simulations used different $c_i$ and $c_j$, and a reason why the experimental results could not achieve 100% accuracy might be the violation of this assumption. Despite this, our method still performs better than others. We will revise our proof and experiments.
>
> $k_i^{x_i}(1-k_i)^{1-x_i}(\frac{1}{1+e^{-(c_j + b_{ji} x_i)}})^{x_j}(1-\frac{1}{1+e^{-(c_j + b_{ji} x_i)}})^{1-x_j} $   (9)
>
> $k_j^{x_j}(1-k_j)^{1-x_j}(\frac{1}{1+e^{-(c_i + b_{ij} x_j)}})^{x_i}(1-\frac{1}{1+e^{-(c_i + b_{ij} x_j)}})^{1-x_i} $  (10)
>
> Q4: On page 2, what do you mean by "The output causal structure here may fall into the solutions up to the ground truth's skeleton"? Similarly, on page 9, what do you mean by "an optimization method may easily fall into its skeleton solutions" and what is the "sub-optimality" you are referring to?
>
> A4: The “sub-optimality” means that the estimated skeleton is consistent with the ground-truth skeleton; while the estimated causal structure with causal directions may not be consistent with the ground-truth one.
>
> Q5: On page 7, the authors relate their reasoning to the proof of Theorem 28 in Peters et al. (2014). I do not think it is necessary to copy the text of Remark 30 from Peters et al. (2014), especially since it is hard to understand without context.
>
> Q5: Thanks for your advice. We will replace Remark 30 with its intuitive description.

---

> > ### Author Response · Authors · 2021-12-04
> > **Response to Reviewer eCeE (Part II)**
> >
> > Q6: On page 10, in the Experiments section, the authors mention that they take the instance of a PDAG returned by PC for evaluation. Which instance do they use exactly? Depending on how the evaluation is performed, one instance might perform better than another. It is also not exactly clear to me how the performance is being evaluated. Are the precision, recall, and F1 score computed in terms of the skeleton, that is, having the corrected edges, or do you also consider the edgemarks? When is a discovery correct and when is it wrong?
> >
> > A6: Since the comparison method PC may return a PDAG, we return one possible DAG for the evaluation. The implementation code for this procedure can be found in the pgmpy python library. For the evaluation of performances, we used the estimated DAG, i.e., the precision, recall, and F1 scores were computed in terms of both edges and edgemarks. Only when the edge with its causal direction was estimated correctly, will it be considered as a correct discovery. We will clarify it.
> >
> > Q7: In the introduction, the categorization of causal discovery algorithms into constraint-based and score-based is introduced twice (on page 1, and then again at the beginning of page 2).
> >
> > A7: The first categorization is introduced for causal discovery algorithms, while the second introduced categorization is mainly and especially for those algorithms with mixed data. We will rephrase the introduction.
> >
> > Q8: Typos, writing errors.
> >
> > A8: Thanks so much for your meticulous review! We will fit them.
> >
> > Q9: What do you mean by "$x_j$'s component with $f_i$" when explaining the underbrace?
> >
> > A9: The underbrace with $x_j$ represents the input component of $f_i$ for the variable $x_j$. We will make it clear.

---

> > > ### Comment · Reviewer_eCeE · 2021-12-10
> > > **Acknowledgment of response and further comments**
> > >
> > > Thank you to the authors for the detailed rebuttal. I still have a few concerns:
> > >
> > > Q2: I still think you could do away with Corollary 1. I would say it is just as easy to show that the "basic non-Gaussian distributions" respect Condition 1 as it is to show that they follow Corollary 1, and I do not see that added benefit of knowing that the Gamma distribution follows S2, whereas the Laplace, Uniform, and Exponential follow S1.
> > >
> > > Q6: If you are using only one possible DAG for the evaluation, then the metrics you mentioned will depend on the specific DAG. Why not average over all DAGs in the equivalence class (PDAG)? I think that would provide a more accurate assessment of performance.
> > >
> > > Further remarks:
> > > Q3: Explicitly adding that assumption would indeed make things much clearer.
> > > Q4: I would just write out what sub-optimality means in that case, that is, use better-known terms like consistency.

---

> > > > ### Author Response · Authors · 2021-12-12
> > > > **Response to Reviewer eCeE**
> > > >
> > > > We greatly appreciate the Reviewer eCeE's comments and suggestions! Please see below our response.
> > > >
> > > > Q2: I still think you could do away with Corollary 1. I would say it is just as easy to show that the "basic non-Gaussian distributions" respect Condition 1 as it is to show that they follow Corollary 1, and I do not see that added benefit of knowing that the Gamma distribution follows S2, whereas the Laplace, Uniform, and Exponential follow S1.
> > > >
> > > > A2: We will show that the "basic non-Gaussian distributions" respect Condition 1 without Corollary 1.
> > > >
> > > > Q6: If you are using only one possible DAG for the evaluation, then the metrics you mentioned will depend on the specific DAG. Why not average over all DAGs in the equivalence class (PDAG)? I think that would provide a more accurate assessment of performance.
> > > >
> > > > A6: We will average all DAGs for the evaluation of PC’s performance.
> > > >
> > > > Q3: Explicitly adding that assumption would indeed make things much clearer.
> > > >
> > > > A3: Yes, we will add this assumption to make things clearer.
> > > >
> > > > Q4: I would just write out what sub-optimality means in that case, that is, use better-known terms like consistency.
> > > >
> > > > A4: We will use “local optima” to represent the estimated structures that belong to the equivalence class of the global optima (the ground truth DAG), and will make it clear in the paper. Thanks!

---

> > > > > ### Comment · Reviewer_eCeE · 2021-12-12
> > > > > **Acknowledgment**
> > > > >
> > > > > Thank you for addressing my final concerns.

---

### Official Review · Reviewer_tEcR · 2021-11-24

**Confidence:** 4
**Overall Score:** 7

**Main Review:**

# Quality and clarity
The paper is not clear, not well-organized, and very hard to read, and it seems to be missing some important points.
- The aim of this study is not clearly stated in Abstract or Introduction. The aim of this study seems extending the studies by Li and Shimizu (2018) and  Wei et al. (2018). However, it is not clear in Abstract or Introduction. In addition, there is no explanation about what is the gap in Li and Shimizu (2018) and Wei et al. (2018).
- The details about the models of Li and Shimizu (2018) and  Wei et al. (2018) can be found in Section 2 (Model definition), but should be moved to Section 1 (Introduction) or an additional section about related studies.
- There is no definition about the probabilistic distribution of $e$ in Definition 1.
- The subscript of function $f$ is abused in Definition 1. $f_c$ and $f_d$ refer to the functions of continuous variables and discrete variables respectively. However, the subscript of $f_i$ corresponds to the index of $x_i$.
- $c$ is also abused: the subscript of $f_c$ means "continuous", and variable $c$ in "$bx+e+c$" means the intercept.
- The last sentence in page 3 says "Now we characterize the condition about the non-Gaussian distributions.", but page 4 says "it’s noteworthy that basic non-Gaussian distributions satisfy the Condition 1". Therefore, it is hard to see if the authors are saying that all the non-Gaussian variables satisfy Condition 1 or not. In addition, it is not clear what "basic non-Gaussian distributions" means.
- The authors "assume that the two models give the same joint distribution of observed variables $x_i$ and $x_j$." in page 5, but there is no explanation about why it does not lose generality.
- The proof (ii) of Theorem 2 aims to prove the identifiability between discrete variables. However, it is not clear how the causal direction is determined by the proof.
- In Equation (11), $e_j$ is limited to follow Logistic(0,1). However, such definition cannot be found in Definition 1.
- There is no explanation about "Notears" in page 10.
- It is hard to evaluate the result shown in Figure 3 because there is no information about the ground truth. In addition, the experiment on the real-world data does not include comparative evaluation with previous methods.

# Originality and significance
As I wrote above, the originality and the significance of this study is not clearly stated in this paper. The contribution of this study seems that the study extended the work of Li and Shimizu (2018) and Wei et al. (2018). However, the significance of it is still unclear.


**Summary:**

This paper studies the identifiability of the causal relationships between continuous and discrete variables.

---

> ### Author Response · Authors · 2021-12-04
> **Response to Reviewer tEcR (Part I)**
>
> We would like to thank the reviewer tEcR for giving helpful comments. We will address the question point by point below.
>
> Q1: The aim of this study is not clearly stated in Abstract or Introduction. The aim of this study seems extending the studies by Li and Shimizu (2018) and Wei et al. (2018). However, it is not clear in Abstract or Introduction. In addition, there is no explanation about what is the gap in Li and Shimizu (2018) and Wei et al. (2018).
>
> A1: The aim of this study is to provide identifiability guarantees for causal discovery with mixed variables. Li and Shimizu (2018) proposed a causal model that consisted of both continuous and discrete variables. Their experiment based on artificial data empirically implied that the model is identifiable, but any mathematical proof has not been given. Wei et al. (2018) developed a mixed causal model and proved its identifiability only in the bivariate cases. In their bivariate identifiability proof, they assumed the noises of continuous variables followed the Laplace distributions and the intercepts for discrete variables are zeros, i.e., $c_i = 0$ in Eq.(2).
> Thus, we aim at enriching the identifiability results for causal discovery with mixed variables, i.e., allowing the proposed model in the proof to have more non-Gaussian distributions to be followed by the noises of continuous variables and possibly nonzero intercepts $c_i$ in Eq.(2), and proving the identifiability in both bivariate and multivariate cases. Please see the last three paragraphs in the Introduction. Thanks for your feedback. We will make them clearer in the Introduction.
>
> Q2: The details about the models of Li and Shimizu (2018) and Wei et al. (2018) can be found in Section 2 (Model definition), but should be moved to Section 1 (Introduction) or an additional section about related studies.
>
> A2: Compared with the models of Li and Shimizu (2018) and Wei et al. (2018), our model is additionally described using an integrated framework shown in Eq. (3) for the concise representation in the identifiability proof. So, we think it is better to move the Model definition to an additional section.
>
> Q3: There is no definition about the probabilistic distribution of $e$ in Definition 1.
>
> A3: From Assumptions A2 and A3, the probabilistic distributions of errors $e_i$ are defined as non-Gaussian and Logistic distributions for continuous and discrete variables, respectively. Please see Definition 1 and the corresponding assumptions A1-A3. We will make it clear in Eq. (3).
>
> Q4: The subscript of function $f$ is abused in Definition 1. $f_c$ and $f_d$ refer to the functions of continuous variables and discrete variables respectively. However, the subscript of $f_i$ corresponds to the index of $x_i$. $c$ is also abused: the subscript of $f_c$ means "continuous", and variable $c$ in "$bx+e+c$" means the intercept.
>
> A4: Thanks for your review. We will replace $f_c, f_d, P_c$ and $P_d$ with $f^{con}, f^{dis}, P^{con}$ and $P^{dis}$ using superscripts.
>
> Q5: The last sentence in page 3 says "Now we characterize the condition about the non-Gaussian distributions.", but page 4 says "it’s noteworthy that basic non-Gaussian distributions satisfy the Condition 1". Therefore, it is hard to see if the authors are saying that all the non-Gaussian variables satisfy Condition 1 or not. In addition, it is not clear what "basic non-Gaussian distributions" means.
>
> A5: Thanks for pointing out this issue. We attempted to characterize Condition 1 that non-Gaussian distributions in our model need to follow so that the identifiability holds. “Basic non-Gaussian distributions” mean those distributions which are commonly used in non-Gaussian data analysis methods including independent component analysis and non-Gaussian causal discovery. We will clarify them.
>
> Q6: The authors "assume that the two models give the same joint distribution of observed variables $x_i$ and $x_j$." in page 5, but there is no explanation about why it does not lose generality.
>
> A6: In the proof of Theorem 2, we’d like to prove that under some conditions, the model for bivariate variables is identifiable (The two models in causal and anti-causal directions entail different joint distributions). Thus, we prove it by the contradictory method, i.e., assuming two models give the same joint distribution, then some conclusion would be drawn to contradict our model’s conditions. We will make it clear at the beginning of the proof.
>
> Q7: The proof (ii) of Theorem 2 aims to prove the identifiability between discrete variables. However, it is not clear how the causal direction is determined by the proof.
>
> A7: Theorem 2 demonstrates the identifiability of our model in the bivariate case, which does not focus on the procedure on how to determine the direction. To know how the causal direction is determined, please see Section 4 Optimization method.

---

> > ### Author Response · Authors · 2021-12-04
> > **Response to Reviewer tEcR (Part II)**
> >
> > Q8: In Equation (11), $e_j$ is limited to follow Logistic(0,1). However, such definition cannot be found in Definition 1.
> >
> > A8: In Definition 1, the LiM model is defined to satisfy Assumptions A1-A3. Please see the Assumption A3, where the distribution of $e_j$ is assumed to follow Logistic(0,1) in Eq.(2).
> >
> > Q9: There is no explanation about "Notears" in page 10.
> >
> > A9: Thanks! We will add an explanation about the compared method “Notears”.
> >
> > Q10: It is hard to evaluate the result shown in Figure 3 because there is no information about the ground truth. In addition, the experiment on the real-world data does not include comparative evaluation with previous methods.
> >
> > A10: It is commonplace for us to encounter the fact that the datasets do not have ground truths. Though there is no public open ground truth towards this dataset, we could use expert or domain knowledge as a reference for evaluation. Many scholars have employed this dataset to verify their methods’ performance (Zhang et al. 2011; Margaritis, 2005). Specifically, in our real-world experiment, the finding “there is no direct link between NOX and MED but they are dependent through intermediate causal relationships” is consistent with the one in Margaritis, (2005); the finding that “TAX, which reflects the government’s housing policy, influences IND, LST, and CRI” is consistent with the one summarized in Kenyon et al (2012). In addition, we employed comparison methods to the real-world dataset. Overall, we see that the logistic method estimated more than 40 edges, and LiNGAM estimated nearly 20 edges, which have more spurious edges than others; laplace and PC methods estimated the TAX as an effect variable, which was not consistent with our common understanding (Kenyon et al.,2012). Due to the large networks and limited space, we will present the results in Supplementary Materials.
> >
> > [Kenyon et al., 2012] Kenyon D A, Langley A H, Paquin B P. Rethinking property tax incentives for business[M]. Cambridge, MA: Lincoln Institute of Land Policy, 2012.

---

### Decision · Program_Chairs · 2022-01-12

**Decision:**

Accept (Poster)

**Comment:**

The paper tackles the problem of causal discovery from mixed data, i.e. when continuous and discrete variables may be present, for both bivariate as well as multivariate settings.
It introduces the so-called Linear Mixed (LiM) model and proves full identifiability under certain assumptions, including causal sufficiency, linear continuous functions with additive non-Gaussian noise, and a linear logistic model for the discrete variables. It also describes the associated LiM algorithm that can find this model, consisting of a global search optimisation phase (based on the so-called quadratic penalty method), followed by a local likelihood optimisation phase. The experimental evaluation shows that, when the assumptions hold, the method compares favourably to other approaches that are not designed to handle this setting.

Reviewers were initially fairly critical of the paper, in particular on clarity of the presentation and several errors / omissions in the text. However the authors made an excellent attempt at answering the points raised, clarifying some and promising to address / resolve others in the final version, so that ultimately all reviewers agreed on recommending acceptance, although insisting presentation could/should still be imporved.

I have to admit on reading the paper I am actually slightly more critical than the reviewers. I fully agree presentation should be improved, but I also notice a tendency to emphasise weaknesses of other methods while focussing on strong points of the proposed method, rather than trying to give a fair and balanced assessment of pros and cons. It is technically ok, but rather niche, relying on strong assumptions where the only ‘real-world’ application (Boston) leads to a questionable output model. Essentially the algorithm is likely to obtain the touted ‘full identifiability’ by overfitting on the data, even though for larger sample sizes the performance actually seems to decrease in some cases. That said, it does at least try to expand existing causal discovery methods to the important mixed data realm, and the proposed approach is new and sound in principle, so despite the caveat above I will follow the reviewers and recommend accept.